# Advances in Microsurgical Treatment Options to Optimize Autologous Free Flap Breast Reconstruction

**DOI:** 10.3390/jcm13195672

**Published:** 2024-09-24

**Authors:** Eric I. Chang

**Affiliations:** The Plastic Surgery Center, The Institute for Advanced Reconstruction, 535 Sycamore Avenue, Shrewsbury, NJ 07702, USA; echangmd@tpscnj.com

**Keywords:** autologous free flap breast reconstruction, neurotization, DIEP flap, alternative donor sites, vascularized lymph node transfer, lymphovenous bypass

## Abstract

**Introduction**: Reconstructive plastic surgeons have made great strides in the field of breast reconstruction to achieve the best results for patients undergoing treatment for breast cancer. As microsurgical techniques have evolved, these patients can benefit from additional treatment modalities to optimize the results of the reconstruction. Free tissue transfer from alternative donor sites for breast reconstruction is routinely performed, which was not possible in the past. Neurotization is now possible to address the numbness and lack of sensation to the reconstructed breast. For those patients who develop lymphedema of the upper extremity as a result of their breast cancer care, supermicrosurgical options are now available to treat and even to prevent the development of lymphedema. This study presents a narrative review regarding the latest microsurgical advancements in autologous free flap breast reconstruction. **Methods**: A literature review was performed on PubMed with the key words “autologous free flap breast reconstruction”, “deep inferior epigastric perforator flap”, “transverse upper gracilis flap”, “profunda artery perforator flap”, “superior gluteal artery perforator flap”, “inferior gluteal artery perforator flap”, “lumbar artery perforator flap”, “breast neurotization”, “lymphovenous bypass and anastomosis”, and “vascularized lymph node transfer”. Articles that specifically focused on free flap breast reconstruction, breast neurotization, and lymphedema surgery in the setting of breast cancer were evaluated and included in this literature review. **Results**: The literature search yielded a total of 4948 articles which were screened. After the initial screening, 413 articles were reviewed to assess the relevance and applicability to the current study. **Conclusions**: Breast reconstruction has evolved tremendously in recent years to provide the most natural and cosmetically pleasing results for those patients undergoing treatment for breast cancer. As technology and surgical techniques have progressed, breast cancer patients now have many more options, particularly if they are interested in autologous reconstruction. These advancements also provide the possibility of restoring sensibility to the reconstructed breast as well as treating the sequela of lymphedema due to their cancer treatment.

## 1. Introduction

Breast reconstruction has made tremendous advances, starting from simply using implants to transferring pedicled flaps and to performing microvascular free perforator flaps. Certainly, there exist a wide array of reconstructive options which can all yield a natural and esthetically pleasing result for the patient and surgeon. As techniques and technology have progressed, reconstructive microsurgeons have sought to improve the outcomes of breast reconstruction in order to achieve optimal results for this patient population.

The deep inferior epigastric perforator (DIEP) flap has become the most popular free flap due to the reliable anatomy of the pedicle and perforators. However, there are patients who are not candidates for DIEP flap reconstruction due to insufficient volume or prior operations, such as an abdominoplasty. In these situations, alternative donor site options are necessary for those patients who desire autologous tissue reconstruction [1].

In addition to reconstructing the breast mound, novel treatment modalities have emerged to address patient wishes and expectations. For instance, reconstruction of the breast and nipple may not be sufficient as patients are also seeking to restore sensation to the reconstructed breasts. Neurotization has been gaining popularity to address these concerns and to restore the patient’s normal sensitivities [2,3,4].

One of the most dreaded complications associated with the treatment of breast cancer is the development of lymphedema. Breast cancer-related lymphedema (BCRL) is a chronic, debilitating condition due to axillary lymph node dissection and/or radiation therapy. Fortunately, lymphovenous bypasses (LVBPs) or lymphovenous anastomosis (LVA) and vascularized lymph node transfer provide physiologic treatment options to combat lymphedema [5]. In fact, the advent of immediate lymphatic reconstruction (ILR) has been promoted to minimize the development of BCRL altogether [6].

## 2. Methods

A literature review was performed on PubMed with the search terms “autologous free flap breast reconstruction”, “deep inferior epigastric perforator flap”, “transverse upper gracilis flap”, “profunda artery perforator flap”, “superior gluteal artery perforator flap”, “inferior gluteal artery perforator flap”, “lumbar artery perforator flap”, “breast neurotization”, “lymphovenous bypass and anastomosis”, and “vascularized lymph node transfer”. Studies evaluating risk factors, complications, patient-reported outcomes, and patient demographics such as age, ethnicity, medical comorbidities, etc., were excluded. Investigational studies with animal models were also not assessed for the purposes of this study. Only articles written in English were included for evaluation in this narrative review.

## 3. Results

A total of 4948 studies were identified and screened by the author to evaluate the advances in autologous tissue breast reconstruction. After the initial query, 413 articles were reviewed based upon the exclusion criteria. The current narrative review includes the most recent and relevant literature in order to provide a comprehensive overview regarding the latest advancements in microsurgery to optimize breast cancer reconstruction.

## 4. Discussion

### 4.1. Alternative Donor Site Options for Microsurgical Breast Reconstruction

The free DIEP flap is one of the most commonly performed autologous tissue options for breast reconstruction. For those patients who do not have a suitable abdomen, there are a wide variety of other donor sites that can be utilized, including the lower extremity, buttocks, and flanks [7]. Furthermore, if there is insufficient volume with a single donor site, the concept of stacked flaps and bipedicled flaps can be employed to achieve a completely autologous tissue reconstruction without resorting to implants.

Traditionally, the buttock has been the primary alternative donor site for free flap breast reconstruction. The inferior gluteal artery perforator (IGAP) flap and the superior gluteal artery perforator (SGAP) flap have been demonstrated to be safe and reliable with low morbidity [8,9]. A large meta-analysis of 14 studies evaluated 667 SGAP flaps and reported a total flap loss rate of 1%, partial flap loss rate of 1%, re-exploration rate of 5%, and a 12% rate of donor site complications, demonstrating the safety and efficacy of gluteal flaps for breast reconstruction [10]. Although one of the early concerns is the need for two stages in the setting of bilateral mastectomies, this is no longer the case with increased surgeon experience and improvements in technique [11,12,13]. In fact, due to the large amount of soft tissue volume, some reconstructive plastic surgeons have advocated using these flaps as the primary option due to the increased speed of harvest compared to the DIEP flap [14,15]. 

For those patients who have previously undergone an abdominoplasty or failed reconstruction from the abdominal donor site, the lumbar artery perforator (LAP) flap offers another secondary option for autologous breast reconstruction and may also aid in improving the esthetic shape and contour of the waistline and buttocks [16]. Although shaping and contouring may be easier with the LAP flap, this flap is technically more challenging as interposition vein grafts may be necessary to address the short pedicle length or to account for a size mismatch between the donor and recipient veins [17,18]. Additional concerns with the lumbar artery perforator flap are related to donor site morbidity, including prolonged seromas and discomfort, noticeable scars, and lumbar hernias [19,20].

Perhaps the lower extremity is now the optimal secondary donor site for free tissue transfer to the breast [21]. Certainly, the anterolateral thigh (ALT) flap is widely used in reconstructive microsurgery for head and neck defects and lower extremity limb salvage, but it can also be utilized for breast reconstruction. Although the transverse upper gracilis (TUG) myocutaneous flap was the first thigh-based flap described, the profunda artery perforator (PAP) flap has recently become the preferred lower extremity free flap for autologous breast reconstruction [22,23,24,25,26]. 

Although a meta-analysis comparing TUG flaps and PAP flaps showed a higher risk of microvascular complications and re-operations during the initial post-operative period with the former flap, another group reported lower complication rates with positive patient satisfaction in 78 patients undergoing breast reconstruction with TUG flaps [27,28]. A recent study of 86 patients undergoing reconstruction with 116 PAP flaps demonstrated a low risk of complications with no flap losses or incidents of microvascular thrombosis. The authors also performed BREAST-Q analysis and found higher post-operative scores indicating high patient satisfaction and improved quality of life after breast reconstruction with PAP flaps [29]. The same authors conducted a similar study comparing 43 patients undergoing IGAP flap breast reconstruction and 51 patients undergoing autologous reconstruction with PAP flaps; they report a lower risk of complications and secondary revisions with the latter option [30].

With the advent of these various alternative free flap options, further refinements have emerged to augment the volume of the reconstructed breast by using more than one flap to reconstruct a single breast. In the setting of a unilateral breast reconstruction, where the abdominal donor site is available, a bipedicled DIEP remains the primary free flap option [31]. However, if the patient has had previous abdominal surgery through a midline incision, the preferred choice would be to proceed with stacked free DIEP flaps to reconstruct the breast [32,33].

For those patients who are interested in autologous tissue reconstruction after bilateral mastectomies, stacking flaps are necessary in order to achieve sufficient volume for both breasts. Certainly, there are numerous free flap combinations that may be stacked in patients who require four flaps. If the abdomen is available but does not have sufficient volume, stacking DIEP flaps with any of the secondary alternative flap choices has been described [34,35,36].

Furthermore, harvest of multiple free flaps has been shown to be safe and effective in achieving a successful and natural reconstruction with a low rate of complications. Haddock et al. proved that patients undergoing stacked flaps had higher rates of deep venous thrombosis and re-operation but no difference in rates of flap loss in 1070 flaps [37]. A recent meta-analysis stated similar findings that stacked and conjoined flaps were not associated with higher rates of complications, regardless of the donor site [38].

Clearly the development of alternative free flaps, as well as stacked flaps, has tremendously advanced the field of autologous microsurgical breast reconstruction. Patients who were formerly deemed inappropriate candidates for free flaps can now safely and reliably have autologous tissue breast reconstruction without having to resort to implants or undergo multiple stages of revision with fat grafting.

### 4.2. Neurotization of Free Flaps for Breast Reconstruction

As the field of breast reconstruction has advanced, the demands of the patients have also increased. One of the major concerns after undergoing a mastectomy is the lack of sensation and numbness of the chest. Even though reconstruction provides the patient with a breast mound that is esthetically pleasing, they do not feel physical touch, which can be distressful. As such, the next iteration of autologous breast reconstruction focused on not only restoring form but also function in terms of providing sensation to the newly reconstructed breast [39]. 

The concept of neurotization to restore breast sensibility centers upon anastomosing nerves within the flap to recipient nerves within the chest. One study evaluating the use of the lateral cutaneous branch of the fourth intercostal nerve and the anterior cutaneous branch of the third intercostal nerve showed comparable axonal counts in both cases. Although both options are suitable recipient nerves, the lateral intercostal nerve may be preferable to preserve sensation of the native breast skin [40]. Various recipient nerves, as well as conduits such as allografts or autologous nerve grafts, have been described with mixed results [41]. 

Evidently, there remains considerable debate regarding the success of nerve coaptation for breast re-innervation. Two meta-analyses reported major inconsistencies in the literature and stated that re-sensation may return spontaneously, but neurotization may enhance the degree and speed of sensory recovery [42,43]. Santanelli et al. showed spontaneous return of sensation in 30 consecutive patients at 6 and 12 months and advocate against neurotization [44]. On the other hand, a more recent study demonstrated improved sensibility of re-innervated flaps with use of a nerve allograft and conduit [45]. Since nerve regeneration can occur but can also be unpredictable, the question remains whether the return of breast sensibility is truly due to neurotization. Moreover, breast sensibility encompasses a wide range of sensations, including light touch, deep touch, temperature, and proprioception, which also warrant further investigation. Certainly, the return of sensation after neurotizing free flaps in breast reconstruction requires larger studies with longer follow-ups to definitively demonstrate the true benefit of this technique. 

Perhaps the advent of new techniques and products may improve the results of breast re-innervation after mastectomies. The use of nerve grafts and conduits may direct nerve regeneration between the donor and recipient nerves more favorably than an anastomosis alone. Clearly, care should be taken to dissect the sensory component of the nerve and to avoid the motor nerves during DIEP flap elevation. Although neurotization of abdominally based flaps is the most commonly performed option and has been well described in the past, patients undergoing free flap reconstruction from other donor sites are also still candidates in order to restore sensation to the reconstructed breasts [46]. Here, we present our patients who underwent breast reconstruction with neurotization using flaps from the lower extremity.

The first patient is a 43-year-old female presenting with left breast cancer who underwent bilateral skin-sparing mastectomies and immediate reconstruction with bilateral free ALT flaps due to her thin body habitus. The lateral femoral cutaneous nerve is a large, consistent sensory nerve within the anterior lateral thigh region that results in numbness when sacrificed during standard flap elevation (Figure 1). In the setting for breast reconstruction with the need for neurotization, this nerve is ideally suited for co-aptation to the medial or lateral intercostal nerves in order to provide re-sensation to the reconstructed breast.

The second patient is a 61-year-old female presenting with recurrent left breast cancer. Since this was her second cancer diagnosis, the patient required bilateral mastectomies and desired to have autologous tissue reconstruction to minimize any potential complications from her prior radiation treatment. The patient had previously had an abdominoplasty and opted for bilateral TUG flaps. Similarly, a large sensory nerve also courses with the vascular pedicle along the proximal aspect of the gracilis muscle, which can also be utilized as the donor nerve to neurotize the TUG flap (Figure 2).

Both patients underwent successful reconstruction with free tissue transfer, as well as neurotization to the third anterior intercostal nerve. The first patient demonstrated sensation to deep and light touch and proprioception but no sensation to temperature at a follow-up of 22 months, while the second patient reported feeling deep touch at 10 months after surgery.

### 4.3. Microsurgical Treatment Modalities for Breast Cancer-Related Lymphedema

The advancements in microsurgery not only provide patients with options for breast reconstruction but also offer patients hope to combat the complications associated with the treatment for breast cancer. Breast cancer-related lymphedema is one of the most common and long-term complications associated with removal of lymph nodes and/or radiation therapy. Even though the risks of developing lymphedema have decreased with the use of sentinel lymph node biopsies instead of complete axillary dissection, approximately 8–30.3% of patients will still develop BCRL [47,48,49,50].

For those patients who develop lymphedema of the upper extremity, microsurgical treatment options, including vascularized lymph node transfer and lymphovenous bypasses, have been devised to address these issues [51,52,53]. These physiological procedures aim to restore the normal function of the lymphatic system, compared to the excisional modalities which are designed to decrease the bulk of the upper extremity through direct excision or liposuction. In fact, immediate lymphatic reconstruction (ILR) or the lymphatic microsurgical preventive healing approach (LYMPHA) employs the same techniques used for lymphovenous bypasses, potentially to prevent the development of lymphedema altogether in those high-risk patients who are undergoing axillary lymph node dissection and/or radiation treatment [54].

Vascularized lymph node transplantation may be performed in an isolated fashion or simultaneously with autologous breast reconstruction. If the patient is undergoing delayed breast reconstruction, the primary option would be to execute a free DIEP flap where the superficial inguinal lymph nodes, based upon the superficial inferior epigastric vessels or superficial circumflex iliac vessels, can be harvested in continuity with the abdominal flap [55,56]. If the patient has already undergone breast reconstruction, a vascularized lymph node transfer is still possible from a wide variety of donor sites, including the supraclavicular region, the submental region, the mesentery, or the omentum [57,58,59,60,61,62,63]. Although the intraperitoneal donor sites are protected regions, donor site lymphedema is a grave complication that should be minimized with the use of reverse lymphatic mapping in order to identify and preserve the lymph node drainage of the associated limb [64].

As such, lymph node transplantation has emerged as a very promising technique to treat BCRL. Multiple literature reviews have reported the efficacy of vascularized lymph node transfers in improving the lymphedema. Despite the promising results, the greatest shortcomings of these studies are small sample sizes as well as the lack of uniform protocols, patient selection, and standardized monitoring. In addition, the mechanism of action of VLNT remains unclear as to whether improvement in the lymphedema is due to scar contracture release or lymphangiogenesis [65,66,67,68]. However, a recent small study of 14 patients undergoing vascularized lymph node transfer had full thickness skin punch biopsies performed after one year. The authors demonstrated increased numbers of podoplanin positive vessels which new evidence that vascularized lymph node transplantation may promote neolymphangiogenesis [69]. Further studies are obviously necessary to assess the true benefits of this surgical intervention.

The second microsurgical technique for the treatment of BCRL entails connecting the lymphatic vessels to venules in order to allow egress of the lymphatic fluid from the affected upper extremity [70]. The ability to perform a lymphovenous bypass or anastomosis requires the identification of patent lymphatic channels. Traditionally, the use of indocyanine green (ICG) lymphangiography was the primary option to localize the lymphatic vessel [71]. Similarly, the advances in technology have followed the advances in technique where additional imaging modalities, such as high-definition/high-frequency ultrasound, magnetic resonance lymphography (MRL), and photoacoustic imaging (PAI), are now available to identify not only the lymphatic vessels but also necessary venules in order to aid in surgical planning [72,73,74,75]. The use of high-definition ultrasound has gained popularity due to the added ability to identify nearby venules, but it is highly user dependent with a substantial learning curve. MRL provides a three-dimensional image but is expensive and may have limited sensitivity for smaller caliber vessels. 

Previously, this surgical option was only possible in earlier stages of lymphedema where the lymphatic vessels are not sclerosed from chronic fibrosis. As a result, these patients were then recommended to undergo a vascularized lymph node transfer for the treatment of the lymphedema. However, the advent of new imaging methods has enabled microsurgeons to identify lymphatic vessels that were not able to be visualized with ICG lymphography alone. These patients with more advanced stages of lymphedema are now also candidates for LVA prior to resorting to a lymph node transfer [76].

As the field of lymphedema surgery has advanced, reconstructive microsurgeons have employed the use of LVBP for immediate lymphatic reconstruction in the hopes of preventing BCRL. Once again, lymphatic mapping is necessary to identify the lymphatic vessels and lymph nodes draining the upper extremity. After completion of the axillary dissection, the lymphatic vessels may be anastomosed to nearby veins, in a similar fashion to the LVA, to allow continued lymph flow from the affected arm [77]. A recent study demonstrated a 9% overall incidence of lymphedema in 90 patients undergoing ILR after axial clearance [78]. Preliminary results of a randomized controlled study from another institution revealed similar results, with a lymphedema incidence of 9.5% in patients undergoing ILR compared to 32% in patients who did not have reconstruction [79]. However, another study showed no difference in development of breast cancer-related lymphedema in patients who did and did not have LYMPHA (31.1% vs. 33.3%, *p* > 0.99) [80]. Indeed, larger trials with longer follow-up are necessary to determine the true value and benefits from immediate lymphatic reconstruction, but these microsurgical advancements may offer breast cancer patients an opportunity to decrease the risk of lymphedema which was not available in the past.

## 5. Conclusions

Breast reconstruction continues to evolve with improved techniques and technology, and plastic and reconstructive microsurgeons need to keep pace with these advancements. The deep inferior epigastric perforator flap is often the primary option for free flap breast reconstruction, but many other donor sites are available which should be within the armamentarium of the reconstructive plastic surgeon. Neurotization presents another opportunity to achieve as natural a reconstruction as possible and can also be readily applied to other flap options with excellent results. Super-microsurgery for the treatment of lymphedema provides another avenue to optimize results and provide relief for the sequelae of breast cancer care in this patient population.

## Figures and Tables

**Figure 1 jcm-13-05672-f001:**
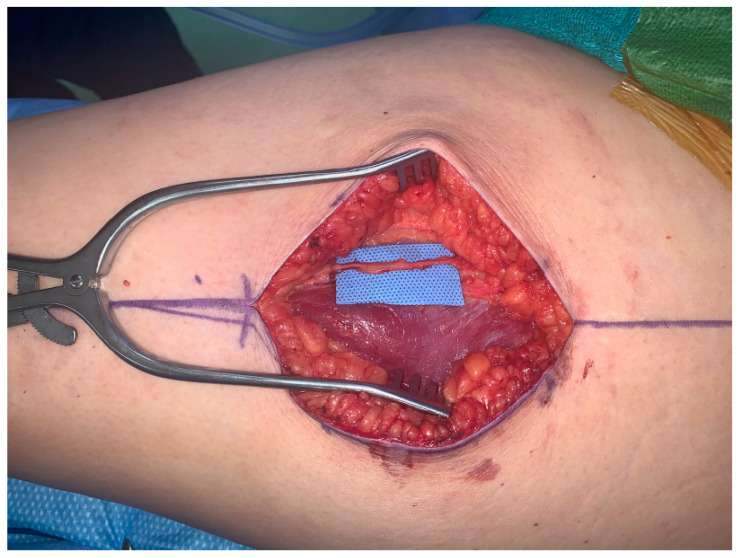
Lateral femoral cutaneous nerve for neurotization of free ALT flap.

**Figure 2 jcm-13-05672-f002:**
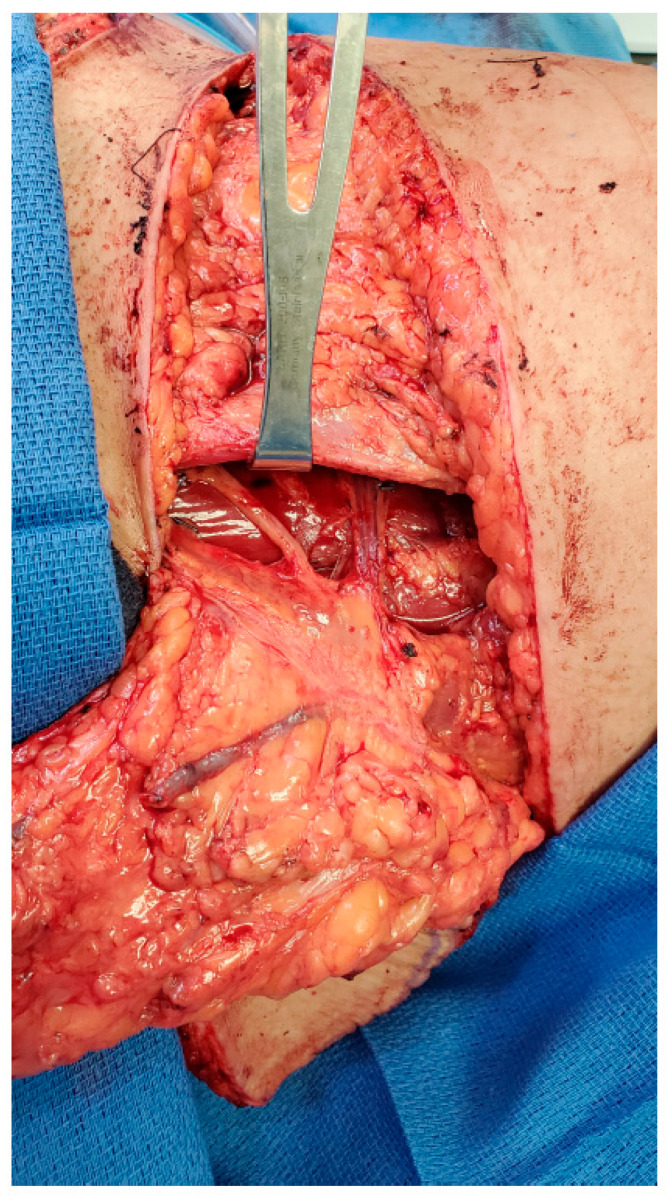
Anterior branch of obturator nerve for neurotization of TUG flap.

## Data Availability

No new data were created with the study.

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
