# Peer review of "Advances in Microsurgical Treatment Options to Optimize Autologous Free Flap Breast Reconstruction"

_jcm, 2024, doi:10.3390/jcm13195672_

Round 1
Reviewer 1 Report
Comments and Suggestions for Authors
Thank you for your submission In general a nice overview, although not really providing the reader with any novel information as such.
It might be recommendable to go more into detail with several of the subjects, not to remain as superficial.
Author Response
Thank you for your submission In general a nice overview, although not really providing the reader with any novel information as such.
It might be recommendable to go more into detail with several of the subjects, not to remain as superficial.
Thank you for the comments. We have added more details to all the subjects in order to provide a more in depth discussion of the various topics included in this review article.
Reviewer 2 Report
Comments and Suggestions for Authors
In this review, the author gives a superficial overview of the role of microsurgery in breast reconstruction, well-pitched for a readership unfamiliar with the topic.
Few specific points should be addressed:
Option of using lipotransfer to augment flap volume should be discussed in addition to stacked flaps and implants.
Evidence for the benefits of neurotization is weak and should be discussed in more detail.
The inclusion of the two case reports of neurotized alt and tug is out of place in a review, and should be omitted.
The unpredictable results of vascularised lymph node transfer and LVA should be discussed.
The text should undergo language editing.
Comments on the Quality of English LanguageThe manuscript contains a few grammatical errors and would benefit from proofreading.
Author Response
In this review, the author gives a superficial overview of the role of microsurgery in breast reconstruction, well-pitched for a readership unfamiliar with the topic.
Few specific points should be addressed:
1. Option of using lipotransfer to augment flap volume should be discussed in addition to stacked flaps and implants.
We thank the reviewer for the comments. We have not included these additional options as we are focusing primarily on microsurgical advances in breast reconstruction. Certainly lipotransfer is a valuable tool as patients have even undergone full breast reconstruction using only fat grafting with the BRAVA technique.
2. Evidence for the benefits of neurotization is weak and should be discussed in more detail.
The reviewer is certainly correct that there is not a clear consensus regarding the benefits of neurotization which has been more thoroughly addressed in the paper.
3. The inclusion of the two case reports of neurotized alt and tug is out of place in a review, and should be omitted.
We truly appreciate the reviewer’s comments but hoped to demonstrate more specifically the new microsurgical innovations in breast reconstruction. We feel the 2 case reports provide more concrete examples to exemplify how the field has progressed.
4. The unpredictable results of vascularised lymph node transfer and LVA should be discussed.
The reviewer has correctly noted that the results from lymphatic surgery can be difficult to quantify. We have included this within in the Discussion with 4 additional references documenting these comments.
5. The text should undergo language editing.
The entire manuscript was reviewers again and proofread for grammar and syntax.
Reviewer 3 Report
Comments and Suggestions for Authors
The author mentions the advancements in microsurgery for breast reconstruction.
While the discussion is well-organized, I believe it is more important to present cases of the most common current breast reconstruction method, the free DIEP flap, as the author suggests. Please provide patient case figures that show the results of breast reconstruction, specifically whether a beautiful breast can be reconstructed. This would be most appropriate to the main focus of this paper, and I do not find significant value in the two figures showing nerve images.
If the author wants to demonstrate a patient treated by neurotization of free flap, please present figures of the reconstructed breast along with information on the benefits the patient gained from the nerve reconstruction.
Regarding the LV bypass discussed in lines 192-199, it does not reflect recent trends. While ICG lymphography is an important option, the identification of lymphatic vessels using ultrasonography is currently a key topic (Akitatsu Hayashi, Giuseppe Visconti). Moreover, the notion that LV bypass is only applicable to early cases has significantly changed. Some groups are performing LVA on degenerated lymphatic vessels, and many groups are achieving good results with LV bypass in severe lymphedema cases (Yukio Seki, Johnson C-S Yang). I believe these facts should be included with referrences.
Author Response
The author mentions the advancements in microsurgery for breast reconstruction.
1. While the discussion is well-organized, I believe it is more important to present cases of the most common current breast reconstruction method, the free DIEP flap, as the author suggests. Please provide patient case figures that show the results of breast reconstruction, specifically whether a beautiful breast can be reconstructed. This would be most appropriate to the main focus of this paper, and I do not find significant value in the two figures showing nerve images
We certainly appreciate the reviewer’s comments and the DIEP flap is clearly the workhorse flap for free flap breast reconstruction. However we hoped to highlight the new advances in microsurgical options for breast reconstruction beyond the DIEP flap.
2. If the author wants to demonstrate a patient treated by neurotization of free flap, please present figures of the reconstructed breast along with information on the benefits the patient gained from the nerve reconstruction.
We thank the reviewer for the comments and the case reports and figures are meant to show more specifically and clearly the novel microsurgical options available to optimize breast reconstruction. We believe the figures help to showcase the technique which is important to advance the field of free flap breast reconstruction. The information regarding the benefits of neurotization have been included in the paper.
3. Regarding the LV bypass discussed in lines 192-199, it does not reflect recent trends. While ICG lymphography is an important option, the identification of lymphatic vessels using ultrasonography is currently a key topic (Akitatsu Hayashi, Giuseppe Visconti). Moreover, the notion that LV bypass is only applicable to early cases has significantly changed. Some groups are performing LVA on degenerated lymphatic vessels, and many groups are achieving good results with LV bypass in severe lymphedema cases (Yukio Seki, Johnson C-S Yang). I believe these facts should be included with referrences.
The reviewer has provided extremely beneficial insights to improve the paper. We have added a more thorough evaluation of imaging techniques to identify lymphatic vessels as well as included information regarding lymphovenous bypasses for advanced stage lymphedema.
Round 2
Reviewer 3 Report
Comments and Suggestions for Authors
The author has appropriately revised the manuscript following the reviewer's suggestions and now considers it to be in an acceptable.
Author Response
I truly appreciate the Reviewer’s insights in helping me to improve upon the paper. Thank you very much for your comments.